# Convex Relaxations for Permutation Problems

**Fajwel Fogel**
C.M.A.P., École Polytechnique,
Palaiseau, France
fogel@cmap.polytechnique.fr

**Rodolphe Jenatton**
CRITEO, Paris & C.M.A.P., École Polytechnique,
Palaiseau, France
jenatton@cmap.polytechnique.fr

**Francis Bach**
INRIA, SIERRA Project-Team & D.I.,
École Normale Supérieure, Paris, France.
francis.bach@ens.fr

**Alexandre d'Aspremont**
CNRS & D.I., UMR 8548,
École Normale Supérieure, Paris, France.
aspremon@ens.fr

## Abstract

Seriation seeks to reconstruct a linear order between variables using unsorted similarity information. It has direct applications in archeology and shotgun gene sequencing for example. We prove the equivalence between the seriation and the combinatorial 2-SUM problem (a quadratic minimization problem over permutations) over a class of similarity matrices. The seriation problem can be solved exactly by a spectral algorithm in the noiseless case and we produce a convex relaxation for the 2-SUM problem to improve the robustness of solutions in a noisy setting. This relaxation also allows us to impose additional structural constraints on the solution, to solve semi-supervised seriation problems. We present numerical experiments on archeological data, Markov chains and gene sequences.

## 1 Introduction

We focus on optimization problems written over the set of permutations. While the relaxation techniques discussed in what follows are applicable to a much more general setting, most of the paper is centered on the *seriation* problem: we are given a similarity matrix between a set of $n$ variables and assume that the variables can be ordered along a chain, where the similarity between variables decreases with their distance within this chain. The seriation problem seeks to reconstruct this linear ordering based on unsorted, possibly noisy, similarity information.

This problem has its roots in archeology [1]. It also has direct applications in e.g. envelope reduction algorithms for sparse linear algebra [2], in identifying interval graphs for scheduling [3], or in shotgun DNA sequencing where a single strand of genetic material is reconstructed from many cloned shorter reads, i.e. small, fully sequenced sections of DNA [4, 5]. With shotgun gene sequencing applications in mind, many references focused on the *Consecutive Ones Problem* (C1P) which seeks to permute the rows of a binary matrix so that all the ones in each column are contiguous. In particular, [3] studied further connections to interval graphs and [6] crucially showed that a solution to C1P can be obtained by solving the seriation problem on the squared data matrix. We refer the reader to [7, 8, 9] for a much more complete survey of applications.

On the algorithmic front, the seriation problem was shown to be NP-Complete by [10]. Archeological examples are usually small scale and earlier references such as [1] used greedy techniques to reorder matrices. Similar techniques were, and are still used to reorder genetic data sets. More general ordering problems were studied extensively in operations research, mostly in connection with the Quadratic Assignment Problem (QAP), for which several convex relaxations were studied in e.g. [11, 12]. Since a matrix is a permutation matrix if and only if it is both orthogonal and

doubly stochastic, much work also focused on producing semidefinite relaxations to orthogonality constraints [13, 14]. These programs are convex hence tractable but the relaxations are usually very large and scale poorly. More recently however, [15] produced a spectral algorithm that exactly solves the seriation problem in a noiseless setting, in results that are very similar to those obtained on the interlacing of eigenvectors for Sturm Liouville operators. They show that for similarity matrices computed from serial variables (for which a total order exists), the ordering of the second eigenvector of the Laplacian (a.k.a. the Fiedler vector) matches that of the variables.

Here, we show that the solution of the seriation problem explicitly minimizes a quadratic function. While this quadratic problem was mentioned explicitly in [15], no connection was made between the combinatorial and spectral solutions. Our result shows in particular that the 2-SUM minimization problem mentioned in [10], and defined below, is polynomially solvable for matrices coming from serial data. This result allows us to write seriation as a quadratic minimization problem over permutation matrices and we then produce convex relaxations for this last problem. This relaxation appears to be more robust to noise than the spectral or combinatorial techniques in a number of examples. Perhaps more importantly, it allows us to impose additional structural constraints to solve semi-supervised seriation problems. We also develop a fast algorithm for projecting on the set of doubly stochastic matrices, which is of independent interest.

The paper is organized as follows. In Section 2, we show a decomposition result for similarity matrices formed from the C1P problem. This decomposition allows to make the connection between the seriation and 2-SUM minimization problems on these matrices. In Section 3 we use these results to write convex relaxations of the seriation problem by relaxing permutation matrices as doubly stochastic matrices in the 2-SUM minimization problem. We also briefly discuss algorithmic and computational complexity issues. Finally Section 4 discusses some applications and numerical experiments.

**Notation.** We write $\mathcal{P}$ the set of permutations of $\{1, \ldots, n\}$. The notation $\pi$ will refer to a permuted vector of $\{1, \ldots, n\}$ while the notation $\Pi$ (in capital letter) will refer to the corresponding matrix permutation, which is a $\{0, 1\}$ matrix such that $\Pi_{ij} = 1$ iff $\pi(j) = i$. For a vector $y \in \mathbb{R}^n$, we write $\mathbf{var}(y)$ its variance, with $\mathbf{var}(y) = \sum_{i=1}^n y_i^2 / n - (\sum_{i=1}^n y_i / n)^2$, we also write $y_{[u,v]} \in \mathbb{R}^{v-u+1}$ the vector $(y_u, \ldots, y_v)^T$. Here, $e_i \in \mathbb{R}^n$ is $i$-th Euclidean basis vector and $\mathbf{1}$ is the vector of ones. We write $\mathbf{S}_n$ the set of symmetric matrices of dimension $n$, $\| \cdot \|_F$ denotes the Frobenius norm and $\lambda_i(X)$ the $i^{\text{th}}$ eigenvalue (in increasing order) of $X$.

## 2 Seriation & consecutive ones

Given a symmetric, binary matrix $A$, we will focus on variations of the following *2-SUM* combinatorial minimization problem, studied in e.g. [10], and written

$$
\begin{array}{ll}
\text{minimize} & \sum_{i,j=1}^n A_{ij}(\pi(i) - \pi(j))^2 \\
\text{subject to} & \pi \in \mathcal{P}.
\end{array}
\tag{1}
$$

This problem is used for example to reduce the envelope of sparse matrices and is shown in [10, Th. 2.2] to be NP-Complete. When $A$ has a specific structure, [15] show that a related matrix reordering problem used for seriation can be solved explicitly by a spectral algorithm. However, the results in [15] do not explicitly link spectral ordering and the optimum of (1). For some instances of $A$ related to seriation and consecutive one problems, we show below that the spectral ordering directly minimizes the objective of problem (1). We first focus on binary matrices, then extend our results to more general unimodal matrices.

### 2.1 Binary matrices

Let $A \in \mathbf{S}_n$ and $y \in \mathbb{R}^n$, we focus on a generalization of the 2-SUM minimization problem

$$
\begin{array}{ll}
\text{minimize} & f(y_\pi) \triangleq \sum_{i,j=1}^n A_{ij}(y_{\pi(i)} - y_{\pi(j)})^2 \\
\text{subject to} & \pi \in \mathcal{P}.
\end{array}
\tag{2}
$$

The main point of this section is to show that if $A$ is the permutation of a similarity matrix formed from serial data, then minimizing (2) recovers the correct variable ordering. We first introduce a few definitions following the terminology in [15].

**Definition 2.1** *We say that the matrix $A \in \mathbf{S}_n$ is an R-matrix (or Robinson matrix) iff it is symmetric and satisfies $A_{i,j} \leq A_{i,j+1}$ and $A_{i+1,j} \leq A_{i,j}$ in the lower triangle, where $1 \leq j < i \leq n$.*

Another way to write the R-matrix conditions is to impose $A_{ij} \leq A_{kl}$ if $|i-j| \leq |k-l|$ off-diagonal, i.e. the coefficients of $A$ decrease as we move away from the diagonal (cf. Figure 1).

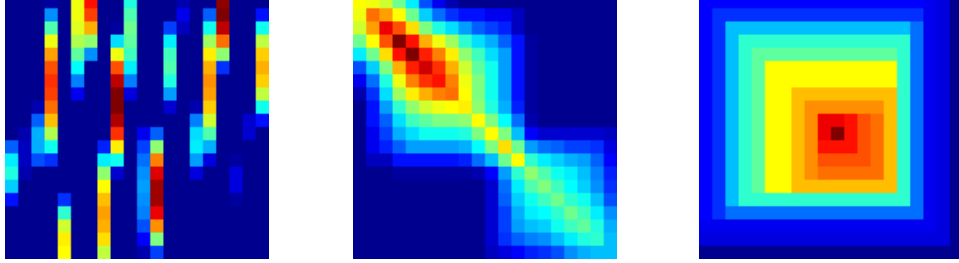

Figure 1: A Q-matrix $A$ (see Def. 2.7), which has unimodal columns *(left)*, its "circular square" $A \circ A^T$ (see Def. 2.8) which is an R-matrix *(center)*, and a matrix $a \circ a^T$ where $a$ is a unimodal vector *(right)*.

**Definition 2.2** *We say that the $\{0,1\}$-matrix $A \in \mathbb{R}^{n \times m}$ is a P-matrix (or Petrie matrix) iff for each column of $A$, the ones form a consecutive sequence.*

As in [15], we will say that $A$ is *pre-R* (resp. *pre-P*) iff there is a permutation $\Pi$ such that $\Pi A \Pi^T$ is an R-matrix (resp. $\Pi A$ is a P-matrix). We now define CUT matrices as follows.

**Definition 2.3** *For $u, v \in [1, n]$, we call $CUT(u, v)$ the matrix such that*

$$CUT(u,v) = \begin{cases} 1 & \text{if } u \leq i, j \leq v \\ 0 & \text{otherwise,} \end{cases}$$

*i.e. $CUT(u, v)$ is symmetric, block diagonal and has one square block equal to one.*

The motivation for this definition is that if $A$ is a $\{0,1\}$ P-matrix, then $AA^T$ is a sum of CUT matrices (with blocks generated by the columns of $A$). This means that we can start by studying problem (2) on CUT matrices. We first show that the objective of (2) has a natural interpretation in this case, as the variance of a subset of $y$ under a uniform probability measure.

**Lemma 2.4** *Let $A = CUT(u, v)$, then $f(y) = \sum_{i,j=1}^{n} A_{ij}(y_i - y_j)^2 = (v - u + 1)^2 \mathbf{var}(y_{[u,v]})$.*

**Proof.** We can write $\sum_{ij} A_{ij}(y_i - y_j)^2 = y^T L_A y$ where $L_A = \mathbf{diag}(A\mathbf{1}) - A$ is the Laplacian of matrix $A$, which is a block matrix equal to $(v - u + 1)\delta_{\{i=j\}} - 1$ for $u \leq i, j \leq v$. ∎

This last lemma shows that solving (2) for CUT matrices amounts to finding a subset of $y$ of size $(u - v + 1)$ with minimum variance. The next lemma characterizes optimal solutions of problem (2) for CUT matrices and shows that its solution splits the coefficients of $y$ in two disjoint intervals.

**Lemma 2.5** *Suppose $A = CUT(u, v)$, and write $z = y_\pi$ the optimal solution to (2). If we call $I = [u, v]$ and $I^c$ its complement in $[1, n]$, then $z_j \notin [\min(z_I), \max(z_I)]$, for all $j \in I^c$, in other words, the coefficients in $z_I$ and $z_{I^c}$ belong to disjoint intervals.*

We can use these last results to show that, at least for some vectors $y$, when $A$ is an R-matrix, then the solution $y_\pi$ to (2) is monotonic.

**Proposition 2.6** *Suppose $C \in \mathbf{S}_n$ is a $\{0,1\}$ pre-R matrix, $A = C^2$, and $y_i = ai + b$ for $i = 1, \ldots, n$ and $a, b \in \mathbb{R}$ with $a \neq 0$. If $\Pi$ is such that $\Pi C \Pi^T$ (hence $\Pi A \Pi^T$) is an R-matrix, then the corresponding permutation $\pi$ solves the combinatorial minimization problem (2) for $A = C^2$.*

**Proof.** Suppose $C$ is $\{0, 1\}$ pre-R, then $C^2$ is pre-R and Lemma 5.2 shows that there exists $\Pi$ such that $\Pi C \Pi^T$ and $\Pi A \Pi^T$ are R-matrices, so we can write $\Pi A \Pi^T$ as a sum of CUT matrices. Furthermore, Lemmas 2.4 and 2.5 show that each CUT term is minimized by a monotonic sequence, but $y_i = ai + b$ means here that all monotonic subsets of $y$ of a given length have the same (minimal) variance, attained by $\Pi y$. So the corresponding $\pi$ also solves problem (2). ∎

## 2.2 Unimodal matrices

Here, based on [6], we first define a generalization of P-matrices called (appropriately enough) Q-matrices, i.e. matrices with unimodal columns. We now show that minimizing (2) also recovers the correct ordering for these more general matrix classes.

**Definition 2.7** *We say that a matrix $A \in \mathbb{R}^{n \times m}$ is a Q-matrix if and only if each column of $A$ is unimodal, i.e. its coefficients increase to a maximum, then decrease.*

Note that R-matrices are symmetric Q-matrices. We call a matrix $A$ *pre-Q* iff there is a permutation $\Pi$ such that $\Pi A$ is a Q-matrix. Next, again based on [6], we define the *circular product* of two matrices.

**Definition 2.8** *Given $A, B^T \in \mathbb{R}^{n \times m}$, and a strictly positive weight vector $w \in \mathbb{R}^m$, their circular product $A \circ B$ is defined as $(A \circ B)_{ij} = \sum_{k=1}^m w_k \min\{A_{ik}, B_{kj}\}$, $i, j = 1, \ldots, n$, note that when $A$ is a symmetric matrix, $A \circ A$ is also symmetric.*

Remark that when $A, B$ are $\{0, 1\}$ matrices and $w = 1$, $\min\{A_{ik}, B_{kj}\} = A_{ik} B_{kj}$, so the circle product matches the regular matrix product $AB^T$. In the appendix we first prove that when $A$ is a Q-matrix, then $A \circ A^T$ is a sum of CUT matrices. This is illustrated in Figure 1.

**Lemma 2.9** *Let $A \in \mathbb{R}^{n \times m}$ a Q-matrix, then $A \circ A^T$ is a conic combination of CUT matrices.*

This last result also shows that $A \circ A^T$ is a R-matrix when $A$ is a Q matrix, as a sum of CUT matrices. These definitions are illustrated in Figure 1. We now recall the central result in [6, Th. 1].

**Theorem 2.10 [6, Th. 1]** *Suppose $A \in \mathbb{R}^{n \times m}$ is pre-Q, then $\Pi A$ is a Q-matrix iff $\Pi(A \circ A^T)\Pi^T$ is a R-matrix.*

We are now ready to show the main result of this section linking permutations which order R-matrices and solutions to problem (2).

**Proposition 2.11** *Suppose $C \in \mathbb{R}^{n \times m}$ is a pre-Q matrix and $y_i = ai + b$ for $i = 1, \ldots, n$ and $a, b \in \mathbb{R}$ with $a \neq 0$. Let $A = C \circ C^T$, if $\Pi$ is such that $\Pi A \Pi^T$ is an R-matrix, then the corresponding permutation $\pi$ solves the combinatorial minimization problem (2).*

**Proof.** If $C \in \mathbb{R}^{n \times m}$ is pre-Q, then Lemma 2.9 and Theorem 2.10 show that there is a permutation $\Pi$ such that $\Pi(C \circ C^T)\Pi^T$ is a sum of CUT matrices (hence a R-matrix). Now as in Propostion 2.6, all monotonic subsets of $y$ of a given length have the same variance, hence Lemmas 2.4 and 2.5 show that $\pi$ solves problem (2). ∎

This result shows that if $A$ is pre-R and can be written $A = C \circ C^T$ with $C$ pre-Q, then the permutation that makes $A$ an R-matrix also solves (2). Since [15] show that sorting the Fiedler vector also orders A as an R-matrix, Prop. 2.11 gives a polynomial time solution to problem (2) when $A = C \circ C^T$ is pre-R with $C$ pre-Q.

## 3 Convex relaxations for permutation problems

In the sections that follow, we will use the combinatorial results derived above to produce convex relaxations of optimization problems written over the set of permutation matrices. Recall that the Fiedler value of a symmetric non negative matrix is the smallest non-zero eigenvalue of its Laplacian. The Fiedler vector is the corresponding eigenvector. We first recall the main result from [15] which shows how to reorder pre-R matrices in a noise free setting.

**Proposition 3.1 [15, Th. 3.3]** *Suppose $A \in \mathbf{S}_n$ is a pre-R-matrix, with a simple Fiedler value whose Fiedler vector $v$ has no repeated values. Suppose that $\Pi \in \mathcal{P}$ is such that the permuted Fiedler vector $\Pi v$ is monotonic, then $\Pi A \Pi^T$ is an R-matrix.*

The results in [15] provide a polynomial time solution to the R-matrix ordering problem in a noise-less setting. While [15] also show how to handle cases where the Fiedler vector is degenerate, these scenarios are highly unlikely to arise in settings where observations on $A$ are noisy and we do not discuss these cases here.

The results in the previous section made the connection between the spectral ordering in [15] and problem (2). In what follows, we will use (2) to produce convex relaxations to matrix ordering problems in a noisy setting. We also show in Section 3 how to incorporate a priori knowledge in the optimization problem. Numerical experiments in Section 4 show that semi-supervised seriation solutions are sometimes significantly more robust to noise than the spectral solutions ordered from the Fiedler vector.

**Permutations and doubly stochastic matrices.** We write $\mathcal{D}_n$ the set of doubly stochastic matrices in $\mathbb{R}^{n \times n}$, i.e. $\mathcal{D}_n = \{ X \in \mathbb{R}^{n \times n} : X \geqslant 0, X\mathbf{1} = \mathbf{1}, X^T\mathbf{1} = \mathbf{1} \}$. Note that $\mathcal{D}_n$ is convex and polyhedral. Classical results show that the set of doubly stochastic matrices is the convex hull of the set of permutation matrices. We also have $\mathcal{P} = \mathcal{D} \cap \mathcal{O}$, i.e. a matrix is a permutation matrix if and only if it is both doubly stochastic and orthogonal. This means that we can directly write a convex relaxation to the combinatorial problem (2) by replacing $\mathcal{P}$ with its convex hull $\mathcal{D}_n$, to get

$$\begin{array}{ll} \text{minimize} & g^T \Pi^T L_A \Pi g \\ \text{subject to} & \Pi \in \mathcal{D}_n, \end{array} \tag{3}$$

where $g = (1, \ldots, n)$. By symmetry, if a vector $\Pi y$ minimizes (3), then the reverse vector also minimizes (3). This often has a significant negative impact on the quality of the relaxation, and we add the linear constraint $e_1^T \Pi g + 1 \leq e_n^T \Pi g$ to break symmetries, which means that we always pick monotonically increasing solutions. Because the Laplacian $L_A$ is always positive semidefinite, problem (3) is a convex quadratic program in the variable $\Pi$ and can be solved efficiently. To provide a solution to the combinatorial problem (2), we then generate permutations from the doubly stochastic optimal solution to (3) (we will describe an efficient procedure to do so in §3).

The results of Section 2 show that the optimal solution to (2) also solves the seriation problem in the noiseless setting when the matrix $A$ is of the form $C \circ C^T$ with $C$ a Q-matrix and $y$ is an affine transform of the vector $(1, \ldots, n)$. These results also hold empirically for small perturbations of the vector $y$ and to improve robustness to noisy observations of $A$, we can average several values of the objective of (3) over these perturbations, solving

$$\begin{array}{ll} \text{minimize} & \mathbf{Tr}(Y^T \Pi^T L_A \Pi Y)/p \\ \text{subject to} & e_1^T \Pi g + 1 \leq e_n^T \Pi g, \Pi\mathbf{1} = \mathbf{1}, \Pi^T\mathbf{1} = \mathbf{1}, \Pi \geq 0, \end{array} \tag{4}$$

in the variable $\Pi \in \mathbb{R}^{n \times n}$, where $Y \in \mathbb{R}^{n \times p}$ is a matrix whose columns are small perturbations of the vector $g = (1, \ldots, n)^T$. Note that the objective of (4) can be rewritten in vector format as $\text{Vec}(\Pi)^T (YY^T \otimes L_A)\text{Vec}(\Pi)/p$. Solving (4) is roughly $p$ times faster than individually solving $p$ versions of (3).

**Regularized convex relaxation.** As the set of permutation matrices $\mathcal{P}$ is the intersection of the set of doubly stochastic matrices $\mathcal{D}$ and the set of orthogonal matrices $\mathcal{O}$, i.e. $\mathcal{P} = \mathcal{D} \cap \mathcal{O}$ we can add a penalty to the objective of the convex relaxed problem (4) to force the solution to get closer to the set of orthogonal matrices.

As a doubly stochastic matrix of Frobenius norm $\sqrt{n}$ is necessarily orthogonal, we would ideally like to solve

$$\begin{array}{ll} \text{minimize} & \frac{1}{p}\mathbf{Tr}(Y^T \Pi^T L_A \Pi Y) - \frac{\mu}{p}\|\Pi\|_F^2 \\ \text{subject to} & e_1^T \Pi g + 1 \leq e_n^T \Pi g, \Pi\mathbf{1} = \mathbf{1}, \Pi^T\mathbf{1} = \mathbf{1}, \Pi \geq 0, \end{array} \tag{5}$$

with $\mu$ large enough to guarantee that the global solution is indeed a permutation. However, this problem is not convex for any $\mu > 0$ since its Hessian is not positive semi-definite (the Hessian $YY^T \otimes L_A - \mu I \otimes I$ is never positive semidefinite when $\mu > 0$ since the first eigenvalue of $L_A$ is 0). Instead, we propose a slightly modified version of (5), which has the same objective function

up to a constant, and is convex for some values of $\mu$. Remember that the Laplacian matrix $L_A$ is always positive semidefinite with at least one eigenvalue equal to zero (strictly one if the graph is connected). Let $P = \mathbf{I} - \frac{1}{n}\mathbf{1}\mathbf{1}^T$.

**Proposition 3.2** *The optimization problem*

$$\begin{aligned} \text{minimize} \quad & \tfrac{1}{p}\mathbf{Tr}(Y^T\Pi^T L_A \Pi Y) - \tfrac{\mu}{p}\|P\Pi\|_F^2 \\ \text{subject to} \quad & e_1^T\Pi g + 1 \le e_n^T\Pi g, \Pi\mathbf{1} = \mathbf{1}, \Pi^T\mathbf{1} = \mathbf{1}, \Pi \ge 0, \end{aligned} \tag{6}$$

*is equivalent to problem* (5) *and their objectives differ by a constant. When* $\mu \le \lambda_2(L_A)\lambda_1(YY^T)$, *this problem is convex.*

**Incorporating structural contraints.** The QP relaxation allows us to add convex structural constraints in the problem. For instance, in archeological applications, one may specify that observation $i$ must appear before observation $j$, i.e. $\pi(i) < \pi(j)$. In gene sequencing applications, one may want to constrain the distance between two elements (e.g. mate reads), which would read $a \le \pi(i) - \pi(j) \le b$ and introduce an affine inequality on the variable $\Pi$ in the QP relaxation of the form $a \le e_i^T\Pi g - e_j^T\Pi g \le b$. Linear constraints could also be extracted from a reference gene sequence. More generally, we can rewrite problem (6) with $n_c$ additional linear constraints as follows

$$\begin{aligned} \text{minimize} \quad & \tfrac{1}{p}\mathbf{Tr}(Y^T\Pi^T L_A \Pi Y) - \tfrac{\mu}{p}\|P\Pi\|_F^2 \\ \text{subject to} \quad & D^T\Pi g + \delta \le 0, \Pi\mathbf{1} = \mathbf{1}, \Pi^T\mathbf{1} = \mathbf{1}, \Pi \ge 0, \end{aligned} \tag{7}$$

where $D$ is a matrix of size $n \times n_c$ and $\delta$ is a vector of size $n_c$. The first column of $D$ is equal to $e_1 - e_n$ and $\delta_1 = 1$ (to break symmetry).

**Sampling permutations from doubly stochastic matrices.** This procedure is based on the fact that a permutation can be defined from a doubly stochastic matrix $D$ by the order induced on a monotonic vector. Suppose we generate a *monotonic* random vector $v$ and compute $Dv$. To each $v$, we can associate a permutation $\Pi$ such that $\Pi Dv$ is monotonically increasing. If $D$ is a permutation matrix, then the permutation $\Pi$ generated by this procedure will be constant, if $D$ is a doubly stochastic matrix but not a permutation, it might fluctuate. Starting from a solution $D$ to problem (6), we can use this procedure to generate many permutation matrices $\Pi$ and we pick the one with lowest cost $y^T\Pi^T L_A \Pi y$ in the combinatorial problem (2). We could also project $\Pi$ on permutations using the Hungarian algorithm, but this proved more costly and less effective.

**Orthogonal relaxation.** Recall that $\mathcal{P} = \mathcal{D} \cap \mathcal{O}$, i.e. a matrix is a permutation matrix if and only if it is both doubly stochastic and orthogonal. So far, we have relaxed the orthogonality constraint to replace it by a penalty on the Frobenius norm. Semidefinite relaxations to orthogonality constraints have been developed in e.g. [12, 13, 14], with excellent approximation bounds, and these could provide alternative relaxation schemes. However, these relaxations form semidefinite programs of dimension $O(n^2)$ (hence have $O(n^4)$ variables) which are out of reach numerically for most of the problems considered here.

**Algorithms.** The convex relaxation in (7) is a quadratic program in the variable $\Pi \in \mathbb{R}^{n \times n}$, which has dimension $n^2$. For reasonable values of $n$ (around a few hundreds), interior point solvers such as MOSEK [17] solve this problem very efficiently. Furthermore, most pre-R matrices formed by squaring pre-Q matrices are very sparse, which considerably speeds up linear algebra. However, first-order methods remain the only alternative beyond a certain scale. We quickly discuss the implementation of two classes of methods: the Frank-Wolfe (a.k.a. conditional gradient) algorithm, and accelerated gradient methods.

Solving (7) using the conditional gradient algorithm in [18] requires minimizing an affine function over the set of doubly stochastic matrices at each iteration. This amounts to solving a classical transportation (or matching) problem for which very efficient solvers exist [19].

On the other hand, solving (7) using accelerated gradient algorithms requires solving a projection step on doubly stochastic matrices at each iteration [20]. Here too, exploiting structure significantly improves the complexity of these steps. Given some matrix $\Pi_0$, the projection problem is written

$$\begin{aligned} \text{minimize} \quad & \tfrac{1}{2}\|\Pi - \Pi_0\|_F^2 \\ \text{subject to} \quad & D^T\Pi g + \delta \le 0, \Pi\mathbf{1} = \mathbf{1}, \Pi^T\mathbf{1} = \mathbf{1}, \Pi \ge 0 \end{aligned} \tag{8}$$

in the variable $\Pi \in \mathbb{R}^{n \times n}$, with parameter $g \in \mathbb{R}^n$. The dual is written

$$
\begin{aligned}
\text{maximize} \quad & -\tfrac{1}{2}\|x\mathbf{1}^T + \mathbf{1}y^T + Dzg^T - Z\|_F^2 - \mathbf{Tr}(Z^T\Pi_0) \\
& + x^T(\Pi_0\mathbf{1} - \mathbf{1}) + y^T(\Pi_0^T\mathbf{1} - \mathbf{1}) + z(D^T\Pi_0 g + \delta) \\
\text{subject to} \quad & z \geq 0, \; Z \geq 0
\end{aligned} \quad (9)
$$

in the variables $Z \in \mathbb{R}^{n \times n}$, $x, y \in \mathbb{R}^n$ and $z \in \mathbb{R}^{n_c}$. The dual is written over decoupled linear constraints in $(z, Z)$ (with $x$ and $y$ are unconstrained). Each subproblem is equivalent to computing a conjugate norm and can be solved in closed form. In particular, the matrix $Z$ is updated at each iteration by $Z = \max\{\mathbf{0}, \; x\mathbf{1}^T + \mathbf{1}y^T + Dzg^T - \Pi_0\}$. Warm-starting provides a significant speed-up. This means that problem (9) can be solved very efficiently by block-coordinate ascent, whose convergence is guaranteed in this setting [21], and a solution to (8) can be reconstructed from the optimum in (9).

## 4 Applications & numerical experiments

**Archeology.** We reorder the rows of the Hodson's Munsingen dataset (as provided by [22] and manually ordered by [6]), to date 59 graves from 70 recovered artifact types (graves from similar periods containing similar artifacts). The results are reported in Table 1 (and in the appendix). We use a fraction of the pairwise orders in [6] to solve the semi-supervised version.

|  | Sol. in [6] | Spectral | QP Reg | QP Reg + 0.1% | QP Reg + 47.5% |
|---|---|---|---|---|---|
| Kendall $\tau$ | 1.00±0.00 | 0.75±0.00 | 0.73±0.22 | 0.76±0.16 | 0.97±0.01 |
| Spearman $\rho$ | 1.00±0.00 | 0.90±0.00 | 0.88±0.19 | 0.91±0.16 | 1.00±0.00 |
| Comb. Obj. | 38520±0 | 38903±0 | 41810±13960 | 43457±23004 | 37602±775 |
| # R-constr. | 1556±0 | 1802±0 | 2021±484 | 2050±747 | 1545±43 |

Table 1: Performance metrics (median and stdev over 100 runs of the QP relaxation, for Kendall's $\tau$, Spearman's $\rho$ ranking correlations (large values are good), the objective value in (2), and the number of R-matrix monotonicity constraint violations (small values are good), comparing Kendall's original solution with that of the Fiedler vector, the seriation QP in (6) and the semi-supervised seriation QP in (7) with 0.1% and 47.5% pairwise ordering constraints specified. Note that the semi-supervised solution actually improves on both Kendall's manual solution and on the spectral ordering.

**Markov chains.** Here, we observe many *disordered* samples from a Markov chain. The mutual information matrix of these variables must be decreasing with $|i - j|$ when ordered according to the true generating Markov chain [23, Th. 2.8.1], hence the mutual information matrix of these variables is a pre-R-matrix. We can thus recover the order of the Markov chain by solving the seriation problem on this matrix. In the following example, we try to recover the order of a Gaussian Markov chain written $X_{i+1} = b_i X_i + \epsilon_i$ with $\epsilon_i \sim N(0, \sigma_i^2)$. The results are presented in Table 2 on 30 variables. We test performance in a noise free setting where we observe the randomly ordered model covariance, in a noisy setting with enough samples (6000) to ensure that the spectral solution stays in a perturbative regime, and finally using much fewer samples (60) so the spectral perturbation condition fails.

**Gene sequencing.** In next generation shotgun gene sequencing experiments, genes are cloned about ten to a hundred times before being decomposed into very small subsequences called "reads", each fifty to a few hundreds base pairs long. Current machines can only accurately sequence these small reads, which must then be reordered by "assembly" algorithms, using the overlaps between reads. We generate artificial sequencing data by (uniformly) sampling reads from chromosome 22 of the human genome from NCBI, then store k-mer hit versus read in a binary matrix (a k-mer is a fixed sequence of k base pairs). If the reads are ordered correctly, this matrix should be C1P, hence we solve the C1P problem on the $\{0, 1\}$-matrix whose rows correspond to k-mers hits for each read, i.e. the element $(i, j)$ of the matrix is equal to one if k-mer $j$ is included in read $i$. This matrix is extremely sparse, as it is approximately band-diagonal with roughly constant degree when reordered appropriately, and computing the Fiedler vector can be done with complexity $O(n \log n)$, as it amounts to computing the second largest eigenvector of $\lambda_n(L)\mathbf{I} - L$, where $L$ is the Laplacian

|         | No noise    | Noise within spectral gap | Large noise  |
|---------|-------------|---------------------------|--------------|
| True    | 1.00±0.00   | 1.00±0.00                 | 1.00±0.00    |
| Spectral| 1.00±0.00   | 0.86±0.14                 | 0.41±0.25    |
| QP Reg  | 0.50±0.34   | 0.58±0.31                 | 0.45±0.27    |
| QP + 0.2% | 0.65±0.29 | 0.40±0.26                 | 0.60±0.27    |
| QP + 4.6% | 0.71±0.08 | 0.70±0.07                 | 0.68±0.08    |
| QP + 54.3% | 0.98±0.01 | 0.97±0.01               | 0.97±0.02    |

Table 2: Kendall's $\tau$ between the true Markov chain ordering, the Fiedler vector, the seriation QP in (6) and the semi-supervised seriation QP in (7) with varying numbers of pairwise orders specified. We observe the (randomly ordered) model covariance matrix (no noise), the sample covariance matrix with enough samples so the error is smaller than half of the spectral gap, then a sample covariance computed using much fewer samples so the spectral perturbation condition fails.

of the matrix. In our experiments, computing the Fiedler vector of a million base pairs sequence takes less than a minute using MATLAB's `eigs` on a standard desktop machine.

In practice, besides sequencing errors (handled relatively well by the high coverage of the reads), there are often repeats in long genomes. If the repeats are longer than the k-mers, the C1P assumption is violated and the order given by the Fiedler vector is not reliable anymore. On the other hand, handling the repeats is possible using the information given by mate reads, i.e. reads that are known to be separated by a given number of base pairs in the original genome. This structural knowledge can be incorporated into the relaxation (7). While our algorithm for solving (7) only scales up to a few thousands base pairs on a regular desktop, it can be used to solve the sequencing problem hierarchically, i.e. to refine the spectral solution. Graph connectivity issues can be solved directly using spectral information.

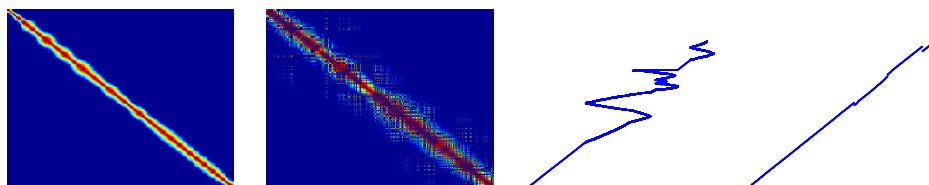

Figure 2: We plot the $reads \times reads$ matrix measuring the number of common k-mers between read pairs, reordered according to the spectral ordering on two regions (two plots on the left), then the Fiedler and Fiedler+QP read orderings versus true ordering (two plots on the right). The semi-supervised solution contains much fewer misplaced reads.

In Figure 2, the two first plots show the result of spectral ordering on simulated reads from human chromosome 22. The full R matrix formed by squaring the reads × kmers matrix is too large to be plotted in MATLAB and we zoom in on two diagonal block submatrices. In the first one, the reordering is good and the matrix has very low bandwidth, the corresponding gene segment (or contig.) is well reconstructed. In the second the reordering is less reliable, and the bandwidth is larger, the reconstructed gene segment contains errors. The last two plots show recovered read position versus true read position for the Fiedler vector and the Fiedler vector followed by semi-supervised seriation, where the QP relaxation is applied to the reads assembled by the spectral solution, on 250 000 reads generated in our experiments. We see that the number of misplaced reads significantly decreases in the semi-supervised seriation solution.

**Acknoledgements.** AA, FF and RJ would like to acknowledge support from a European Research Council starting grant (project SIPA) and a gift from Google. FB would like to acknowledge support from a European Research Council starting grant (project SIERRA). A much more complete version of this paper is available as [16] at arXiv:1306.4805.

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
