[Supplementary Material · Supplementary.pdf]

# 5 Supplementary material

## 5.1 Seriation lemmas

Here, we prove some of the technical lemmas from Section 2.

**Lemma 5.1** *Suppose $A \in \mathbf{S}_n$ is a $\{0,1\}$ R-matrix, and $\Pi A$ is a P-matrix, then $\Pi A \Pi^T$ is an R-matrix.*

**Proof.** Without loss of generality, we can assume that the graph of $A$ is irreducible (otherwise, we simply repeat the proof on each block). If $A \in \mathbf{S}_n$ is an irreducible $\{0,1\}$ R-matrix, then $\mathbf{diag}(A) = \mathbf{1}$. Let $\Pi$ be a permutation such that $\Pi A$ is a P-matrix, so $C = \Pi A \Pi^T$ is a symmetric P-matrix. Let $1 \leq j < i \leq n$, and suppose $C_{ij} = 1$, then $C_{(i-1)j} = 1$ because $C_{jj} = 1$ and $C$ is a P-matrix. Similarly, because $C$ is symmetric, if $C_{ij} = 1$ then $C_{ji} = 1$ and $C_{(j+1)i} = 1$ because $C_{ii} = 1$ and $C$ is a P-matrix, so $C_{i(j+1)} = 1$. This means that $C$ is an R-matrix. ∎

**Lemma 5.2** *Suppose $A \in \mathbf{S}_n$ is a $\{0,1\}$ pre-R matrix, then $\Pi A \Pi^T$ is an R-matrix if and only if $\Pi A^2 \Pi^T$ is an R-matrix.*

**Proof.** If $A \in \mathbf{S}_n$ is a $\{0,1\}$ pre-R matrix, then it must be pre-P (cf. remarks above). [15, Th. 6.3] shows that $\Pi A$ is a P-matrix iff $\Pi A^2 \Pi^T$ is an R-matrix. Combining this with Lemma 5.1 yields the desired result. ∎

Note that a $\{0,1\}$ R-matrix is also a (symmetric) P-matrix. Note also that Lemma 5.1 shows that if $A$ is pre-R, then $\Pi A \Pi^T$ is an R-matrix, hence a P-matrix, and so is $\Pi A$ (it is obtained by permuting the columns of a P-matrix), so $A$ is also pre-P.

**Lemma 5.3** *Let $A \in \mathbf{S}_n$, $y \in \mathbb{R}^n$ and suppose we switch the values of $y_j$ and $y_{j+1}$ calling the new vector $z$, we have*

$$f(y) - f(z) = 4 \sum_{\substack{i=1 \\ i \neq j,\, i \neq j+1}}^{n} \left( \frac{y_j + y_{j+1}}{2} - y_i \right) (y_{j+1} - y_j)(A_{ij+1} - A_{ij})$$

**Proof.** Because $A$ is symmetric, we have

$$
\begin{aligned}
(f(y) - f(z))/2 &= \sum_{\substack{i \neq j,\, i \neq j+1}} A_{ij}(y_i - y_j)^2 + \sum_{\substack{i \neq j,\, i \neq j+1}} A_{ij+1}(y_i - y_{j+1})^2 \\
&\quad - \sum_{\substack{i \neq j,\, i \neq j+1}} A_{ij}(y_i - y_{j+1})^2 - \sum_{\substack{i \neq j,\, i \neq j+1}} A_{ij+1}(y_i - y_j)^2 \\
&= \sum_{\substack{i \neq j,\, i \neq j+1}} 2A_{ij}(y_j - y_{j+1})\left( \frac{y_j + y_{j+1}}{2} - y_i \right) \\
&\quad + \sum_{\substack{i \neq j,\, i \neq j+1}} 2A_{ij+1}(y_{j+1} - y_j)\left( \frac{y_j + y_{j+1}}{2} - y_i \right)
\end{aligned}
$$

which yields the desired result. ∎

**Lemma 5.4** *Suppose $A = CUT(u,v)$, and write $w = y_\pi$ the optimal solution to (2). If we call $I = [u,v]$ and $I^c$ its complement, then*

$$w_j \notin [\min(w_I), \max(w_I)], \quad \text{for all } j \in I^c,$$

*in other words, the coefficients in $w_I$ and $w_{I^c}$ belong to disjoint intervals.*

**Proof.** Without loss of generality, we can assume that the coefficients of $w_I$ are sorted in increasing order. By contradiction, suppose that there is a $w_j$ such that $j \in I^c$ and $w_j \notin [w_u, w_v]$. Suppose also that $w$ is larger than the mean of coefficients inside I, i.e. $w_j \geq \sum_{i=u+1}^{v} w_i / (v - u)$. This, combined with our assumption that $w_j \leq w_v$ and Lemma 5.3 means that switching the values of $w_j$ and $w_v$ will decrease the objective by

$$4 \sum_{i=u}^{v-1} \left( \frac{w_j + w_v}{2} - y_i \right) (w_v - w_j)$$

which is positive by our assumptions on $w_j$ and the mean which contradicts optimality. A symmetric result holds if $w_j$ is smaller than the mean. ■

**Lemma 5.5** *Suppose $A \in \mathbb{R}^{n \times m}$ is a Q-matrix, then $A \circ A^T$ is a conic combination of CUT matrices.*

**Proof.** Suppose, $a \in \mathbb{R}^n$ is a unimodal vector, let us show that the matrix $M = a \circ a^T$ with coefficients $M_{ij} = \min\{a_i, a_j\}$ is a conic combination of CUT matrices. Let $I = \operatorname{argmax}_i a_i$, then $I$ is an index interval $[I_{\min}, I_{\max}]$ because $a$ is unimodal. Call $\bar{a} = \max_i a_i$ and $b = \max_{i \in I^c} a_i$ (with $b = 0$ when $I^c = \emptyset$), the deflated matrix

$$M^- = M - (\bar{a} - b) \, CUT(I_{\min}, I_{\max})$$

can be written $M^- = a^- \circ (a^-)^T$ with

$$a^- = a - (\bar{a} - b)v$$

where $v_i = 1$ iff $i \in I$. By construction $|\operatorname{argmax} M^-| > |I|$, i.e. the size of $\operatorname{argmax} M$ increases by at least one, so this deflation procedure ends after at most $n$ iterations. This shows that $a \circ a^T$ is a conic combination of CUT matrices when $a$ is unimodal. Now, we have $(A \circ A^T)_{ij} = \sum_{k=1}^{n} w_k \min\{A_{ik}, A_{jk}\}$, so $A \circ A^T$ is a sum of $n$ matrices of the form $\min\{A_{ik}, A_{jk}\}$ where each column is unimodal, hence the desired result. ■

## 5.2 Convex relaxation results

We now prove some of the convex relaxation results obtained in Section 3.

**Proposition 5.6** *The optimization problem*

$$\min_{\{\Pi \in \mathcal{D}_n, \, e_1^T \Pi v + 1 \leq e_n^T \Pi v\}} \frac{1}{p} \mathbf{Tr}(Y^T \Pi^T L_A \Pi Y) - \frac{\mu}{p} \|P\Pi\|_F^2 \tag{10}$$

*is equivalent to problem (5), their objectives differ by a constant. Furthermore, when $\mu \leq \lambda_2(L_A)\lambda_1(YY^T)$, this problem is convex.*

**Proof.** Remark that

$$
\begin{aligned}
\|P\Pi\|_F^2 &= \mathbf{Tr}(\Pi^T P^T P \Pi) = \mathbf{Tr}(\Pi^T P \Pi) \\
&= \mathbf{Tr}(\Pi^T (I - \frac{1}{n}\mathbf{1}\mathbf{1}^T)\Pi) = \mathbf{Tr}(\Pi^T \Pi - \frac{1}{n}\mathbf{1}\mathbf{1}^T)) \\
&= \mathbf{Tr}(\Pi^T \Pi) - 1
\end{aligned}
$$

where we used the fact that $P$ is the (symmetric) projector matrix onto the orthogonal of $\mathbf{1}$ and $\Pi$ is doubly stochastic (so $\Pi\mathbf{1} = \Pi^T\mathbf{1} = \mathbf{1}$). We deduce that problem (6) has the same objective function as (5) up to a constant. Moreover, it is convex when $\mu \leq \lambda_2(L_A)$ since the Hessian of the objective is given by

$$\Sigma = \frac{1}{p} YY^T \otimes L_A - \frac{\mu}{p} \cdot \mathbf{I} \otimes P$$

and the eigenvalues of $YY^T \otimes L_A$, which are equal to $\lambda_i(L_A)\lambda_j(YY^T)$ for all $i, j$ in $\{1, \dots, n\}$ are all superior or equal to the eigenvalues of $\mu \cdot \mathbf{I} \otimes P$ which are all smaller than $\mu$. ■

We now show that that minimizing the average of the relaxed problems costs provides in a sense a tighter relaxation to the combinatorial problem 2 than solving individually the relaxed problems.

**Proposition 5.7** *Let $\Pi_0$ be the optimal solution of 2, $\Pi_i^\star$ the optimal solution of 3 with $y = y_i$ and $\Pi_m^\star$ be the optimal solution of 4. $\Pi_0$ is an optimal solution to $\min_{\{\Pi \in \mathcal{P}_n,\, e_1^T \Pi v + 1 \leq e_n^T \Pi v\}} \frac{1}{p} \sum_{i=1}^{p} y_i^T \Pi^T L_A \Pi y_i$ and*

$$\frac{1}{p} \sum_{i=1}^{p} y_i^T \Pi_i^{\star T} L_A \Pi_i^\star y_i \leqslant \frac{1}{p} \sum_{i=1}^{p} y_i^T \Pi_m^{\star T} L_A \Pi_m^\star y_i \leqslant \frac{1}{p} \sum_{i=1}^{p} y_i^T \Pi_0^T L_A \Pi_0 y_i \ .$$

**Proof.** $\Pi_0$ is optimal for all problems 2 with $y = y_i$ so it is optimal for $\min_{\{\Pi \in \mathcal{P}_n,\, e_1^T \Pi v + 1 \leq e_n^T \Pi v\}} \frac{1}{p} \sum_{i=1}^{p} y_i^T \Pi^T L_A \Pi y_i$. The first inequality comes from the optimality of each $\Pi_i^\star$ for problem 3 with $y = y_i$. The second inequality comes from the optimality of $\Pi_m^\star$ for the relaxed problem 4. ∎

With independent constraints ($D$ full rank), at each iteration, the full variable updates in the dual Euclidean projection problem over doubly stochastic matrices are given by

- $Z = \max\{\mathbf{0},\ x\mathbf{1}^T + \mathbf{1}y^T + Dzg^T - \Pi_0\}$
- $x = \frac{1}{n}(\Pi_0 \mathbf{1} - (y^T \mathbf{1} + 1)\mathbf{1} - Dzg^T \mathbf{1} + Z\mathbf{1})$
- $y = \frac{1}{n}(\Pi_0^T \mathbf{1} - (x^T \mathbf{1} + 1)\mathbf{1} + Z^T \mathbf{1})$
- $z = \frac{1}{\|g\|_2^2} \max\{0,\ (D^T D)^{-1}(D^T(Z + \Pi_0)g + \delta - D^T xg^T \mathbf{1})\}$.

The convergence of the algorithm can be monitored through the duality gap formed by the difference of the objective of (8) and (9).

### 5.3 Numerical experiments

|  | **Kendall Sol.** | **Spectral** | **QP** | **QP Reg** | **QP + 0.1%** |
|---|---|---|---|---|---|
| Kendall $\tau$ | 1.00±0.00 | 0.75±0.00 | 0.70±0.22 | 0.73±0.22 | 0.76±0.16 |
| Spearman $\rho$ | 1.00±0.00 | 0.90±0.00 | 0.87±0.19 | 0.88±0.19 | 0.91±0.16 |
| Comb. Obj. | 38520±0 | 38903±0 | 42293±14928 | 41810±13960 | 43457±23004 |
| # R-constr. | 1556±0 | 1802±0 | 2029±491 | 2021±484 | 2050±747 |
|  | **QP + 0.2%** | **QP + 0.5%** | **QP + 1.1%** | **QP + 2.4%** | **QP + 5.1%** |
| Kendall $\tau$ | 0.79±0.07 | 0.80±0.04 | 0.81±0.03 | 0.83±0.03 | 0.86±0.02 |
| Spearman $\rho$ | 0.93±0.05 | 0.94±0.03 | 0.94±0.02 | 0.96±0.02 | 0.97±0.01 |
| Comb. Obj. | 43227±12475 | 44970±8456 | 43748±7989 | 43064±8105 | 42575±5779 |
| # R-constr. | 2026±485 | 2116±377 | 2045±356 | 2026±358 | 1978±288 |
|  | **QP + 10.7%** | **QP + 22.3%** | **QP + 47.5%** | **QP + 100%** |  |
| Kendall $\tau$ | 0.89±0.02 | 0.93±0.01 | 0.97±0.01 | 0.99±0.00 |  |
| Spearman $\rho$ | 0.98±0.01 | 0.99±0.00 | 1.00±0.00 | 1.00±0.00 |  |
| Comb. Obj. | 40452±4107 | 38126±1916 | 37602±775 | 37203±125 |  |
| # R-constr. | 1855±191 | 1646±110 | 1545±43 | 1512±9 |  |

Table 3: Performance metrics (median and stdev over 100 runs of the QP relaxation, for Kendall's $\tau$, Spearman's $\rho$, the objective value in (1), and the number of R-matrix monotonicity constraint violations), comparing Kendall's original solution with that of the Fiedler vector, the seriation QP in (6) and the semi-supervised seriation QP in (7) with an increasing number of pairwise ordering constraints specified, out of the 3422 possible pairs in this problem. Note that the semi-supervised solution actually improves on Kendall's original solution.

|  | **Kendall Sol.** | **Spectral** | **QP** | **QP Reg** | **QP + 0.1%** |
|---|---|---|---|---|---|
| **Kendall $\tau$** | 1.00 | 0.76 | 0.86 | 0.89 | 0.86 |
| **Spearman $\rho$** | 1.00 | 0.90 | 0.96 | 0.97 | 0.97 |
| **Comb Obj.** | 38520.00 | 38903.00 | 30862.00 | 31369.00 | 32464.00 |
| **# R-constr.** | 1556.00 | 1802.00 | 1335.00 | 1371.00 | 1465.00 |
|  | **QP + 0.2%** | **QP + 0.5%** | **QP + 1.1%** | **QP + 2.4%** | **QP + 5.1%** |
| **Kendall $\tau$** | 0.86 | 0.87 | 0.90 | 0.89 | 0.90 |
| **Spearman $\rho$** | 0.96 | 0.97 | 0.98 | 0.98 | 0.98 |
| **Comb Obj.** | 31082.00 | 32345.00 | 32956.00 | 32209.00 | 33669.00 |
| **# R-constr.** | 1361.00 | 1480.00 | 1514.00 | 1460.00 | 1559.00 |
|  | **QP + 10.7%** | **QP + 22.3%** | **QP + 47.5%** | **QP + 100%** |  |
| **Kendall $\tau$** | 0.92 | 0.96 | 0.99 | 1.00 |  |
| **Spearman $\rho$** | 0.99 | 1.00 | 1.00 | 1.00 |  |
| **Comb Obj.** | 34303.00 | 33731.00 | 35270.00 | 36758.00 |  |
| **# R-constr.** | 1561.00 | 1456.00 | 1461.00 | 1492.00 |  |

Table 4: Performance metrics (best of 100 runs of the QP relaxation, for Kendall's $\tau$, Spearman's $\rho$, the objective value in (1), and the number of R-matrix monotonicity constraint violations), comparing Kendall's original solution with that of the Fiedler vector, the seriation QP in (6) and the semi-supervised seriation QP in (7) with an increasing number of pairwise ordering constraints specified, out of the 3422 possible pairs in this problem. Note that the semi-supervised solution actually improves on Kendall's original solution.

|  | **No noise** | **Noise within spectral gap** | **Large noise** |
|---|---|---|---|
| **True** | 1.00±0.00 | 1.00±0.00 | 1.00±0.00 |
| **Spectral** | 1.00±0.00 | 0.96±0.10 | 0.46±0.29 |
| **QP** | 0.57±0.36 | 0.52±0.35 | 0.42±0.32 |
| **QP Reg** | 0.66±0.36 | 0.56±0.35 | 0.39±0.32 |
| **QP Reg + 0.1%** | 0.77±0.33 | 0.38±0.32 | 0.77±0.33 |
| **QP Reg + 0.7%** | 0.80±0.25 | 0.78±0.29 | 0.80±0.27 |
| **QP Reg + 1.4%** | 0.80±0.23 | 0.78±0.25 | 0.79±0.20 |
| **QP Reg + 2.5%** | 0.83±0.13 | 0.83±0.12 | 0.81±0.10 |
| **QP Reg + 4.6%** | 0.87±0.07 | 0.86±0.06 | 0.85±0.09 |
| **QP Reg + 8.7%** | 0.91±0.04 | 0.90±0.04 | 0.90±0.04 |
| **QP Reg + 16.1%** | 0.95±0.02 | 0.95±0.02 | 0.94±0.02 |
| **QP Reg + 29.7%** | 0.98±0.01 | 0.98±0.01 | 0.98±0.01 |
| **QP Reg + 54.3%** | 1.00±0.00 | 1.00±0.00 | 1.00±0.00 |
| **QP Reg + 100.0%** | 1.00±0.00 | 1.00±0.00 | 1.00±0.00 |

Table 5: Median $\pm$ stdev. on Spearman's $\rho$ between the true Markov chain ordering, the Fiedler vector, the seriation QP in (6) and the semi-supervised seriation QP in (7) with varying numbers of pairwise orders specified. We observe the (randomly ordered) model covariance matrix (no noise), the sample covariance matrix with enough samples so the error is smaller than half of the spectral gap, then much fewer samples (Large noise).

|  | No noise | Noise within spectral gap | Large noise |
|---|---|---|---|
| **True** | 1.00 | 1.00 | 1.00 |
| **Spectral** | 1.00 | 1.00 | 1.00 |
| **QP** | 1.00 | 1.00 | 1.00 |
| **QP Reg** | 1.00 | 1.00 | 1.00 |
| **QP Reg add 0.1% cons** | 1.00 | 0.99 | 1.00 |
| **QP Reg add 0.7% cons** | 0.97 | 0.98 | 0.96 |
| **QP Reg add 1.4% cons** | 0.94 | 0.95 | 0.97 |
| **QP Reg add 2.5% cons** | 0.98 | 0.95 | 0.95 |
| **QP Reg add 4.6% cons** | 0.95 | 0.97 | 0.95 |
| **QP Reg add 8.7% cons** | 0.98 | 0.99 | 0.96 |
| **QP Reg add 16.1% cons** | 0.99 | 0.98 | 0.99 |
| **QP Reg add 29.7% cons** | 1.00 | 1.00 | 1.00 |
| **QP Reg add 54.3% cons** | 1.00 | 1.00 | 1.00 |
| **QP Reg add 100.0% cons** | 1.00 | 1.00 | 1.00 |

Table 6: Best Spearman's $\rho$ between the true Markov chain ordering, the Fiedler vector, the seriation QP in (6) and the semi-supervised seriation QP in (7) with varying numbers of pairwise orders specified. We observe the (randomly ordered) model covariance matrix (no noise), the sample covariance matrix with enough samples so the error is smaller than half of the spectral gap, then much fewer samples (Large noise).

|  | No noise | Noise within spectral gap | Large noise |
|---|---|---|---|
| **True** | 1.00±0.00 | 1.00±0.00 | 1.00±0.00 |
| **Spectral** | 1.00±0.00 | 0.86±0.14 | 0.41±0.25 |
| **QP** | 0.49±0.34 | 0.55±0.31 | 0.45±0.27 |
| **QP Reg** | 0.50±0.34 | 0.58±0.31 | 0.45±0.27 |
| **QP Reg + 0.1%** | 0.65±0.29 | 0.40±0.26 | 0.60±0.27 |
| **QP Reg + 0.7%** | 0.66±0.21 | 0.65±0.23 | 0.62±0.23 |
| **QP Reg + 1.4%** | 0.66±0.19 | 0.63±0.21 | 0.65±0.17 |
| **QP Reg + 2.5%** | 0.67±0.12 | 0.66±0.11 | 0.65±0.10 |
| **QP Reg + 4.6%** | 0.71±0.08 | 0.70±0.07 | 0.68±0.08 |
| **QP Reg + 8.7%** | 0.75±0.05 | 0.75±0.06 | 0.75±0.05 |
| **QP Reg + 16.1%** | 0.83±0.05 | 0.83±0.05 | 0.82±0.05 |
| **QP Reg + 29.7%** | 0.92±0.03 | 0.91±0.03 | 0.91±0.03 |
| **QP Reg + 54.3%** | 0.98±0.01 | 0.97±0.01 | 0.97±0.02 |
| **QP Reg + 100.0%** | 1.00±0.00 | 1.00±0.00 | 0.99±0.00 |

Table 7: Median ± stdev. on Kendall's $\tau$ between the true Markov chain ordering, the Fiedler vector, the seriation QP in (6) and the semi-supervised seriation QP in (7) with varying numbers of pairwise orders specified. We observe the (randomly ordered) model covariance matrix (no noise), the sample covariance matrix with enough samples so the error is smaller than half of the spectral gap, then much fewer samples (Large noise).

| | No noise | Noise within spectral gap | Large noise |
|---|---|---|---|
| **True** | 1.00 | 1.00 | 1.00 |
| **Spectral** | 1.00 | 0.99 | 0.98 |
| **QP** | 1.00 | 0.97 | 0.98 |
| **QP Reg** | 1.00 | 0.97 | 0.97 |
| **QP Reg add 0.1% cons** | 0.98 | 0.95 | 0.97 |
| **QP Reg add 0.7% cons** | 0.89 | 0.94 | 0.88 |
| **QP Reg add 1.4% cons** | 0.85 | 0.85 | 0.91 |
| **QP Reg add 2.5% cons** | 0.91 | 0.86 | 0.83 |
| **QP Reg add 4.6% cons** | 0.83 | 0.89 | 0.85 |
| **QP Reg add 8.7% cons** | 0.91 | 0.92 | 0.86 |
| **QP Reg add 16.1% cons** | 0.95 | 0.93 | 0.94 |
| **QP Reg add 29.7% cons** | 0.99 | 0.98 | 0.98 |
| **QP Reg add 54.3% cons** | 1.00 | 1.00 | 1.00 |
| **QP Reg add 100.0% cons** | 1.00 | 1.00 | 1.00 |

Table 8: Best Kendall's $\tau$ between the true Markov chain ordering, the Fiedler vector, the seriation QP in (6) and the semi-supervised seriation QP in (7) with varying numbers of pairwise orders specified. We observe the (randomly ordered) model covariance matrix (no noise), the sample covariance matrix with enough samples so the error is smaller than half of the spectral gap, then much fewer samples (Large noise).

| | No noise | Noise within spectral gap | Large noise |
|---|---|---|---|
| **True** | 0±0 | 142±99 | 823±250 |
| **Spectral** | 0±0 | 780±528 | 1715±560 |
| **QP** | 1782±917 | 1640±754 | 1746±459 |
| **QP Reg** | 1766±919 | 1566±746 | 1734±455 |
| **QP Reg + 0.1%** | 1738±690 | 2035±596 | 1942±442 |
| **QP Reg + 0.7%** | 1886±535 | 1998±529 | 2164±392 |
| **QP Reg + 1.4%** | 1982±546 | 2160±476 | 2308±393 |
| **QP Reg + 2.5%** | 1948±484 | 2048±430 | 2352±364 |
| **QP Reg + 4.6%** | 1818±381 | 1934±391 | 2246±368 |
| **QP Reg + 8.7%** | 1660±325 | 1757±318 | 2105±319 |
| **QP Reg + 16.1%** | 1157±307 | 1279±329 | 1740±360 |
| **QP Reg + 29.7%** | 547±225 | 780±264 | 1278±305 |
| **QP Reg + 54.3%** | 150±100 | 322±130 | 932±275 |
| **QP Reg + 100.0%** | 0±0 | 142±99 | 798±251 |

Table 9: Median ± stdev. on number of violated R-matrix monotonicity constraints for the true Markov chain ordering, the Fiedler vector, the seriation QP in (6) and the semi-supervised seriation QP in (7) with varying numbers of pairwise orders specified. We observe the (randomly ordered) model covariance matrix (no noise), the sample covariance matrix with enough samples so the error is smaller than half of the spectral gap, then much fewer samples (Large noise).

|  | No noise | Noise within spectral gap | Large noise |
|---|---|---|---|
| **True** | 0 | 23 | 352 |
| **Spectral** | 0 | 89 | 414 |
| **QP** | 0 | 261 | 488 |
| **QP Reg** | 0 | 263 | 547 |
| **QP Reg add 0.1% cons** | 115 | 439 | 916 |
| **QP Reg add 0.7% cons** | 655 | 456 | 1060 |
| **QP Reg add 1.4% cons** | 822 | 1064 | 1139 |
| **QP Reg add 2.5% cons** | 587 | 1169 | 1555 |
| **QP Reg add 4.6% cons** | 1002 | 902 | 1507 |
| **QP Reg add 8.7% cons** | 750 | 710 | 1379 |
| **QP Reg add 16.1% cons** | 336 | 534 | 997 |
| **QP Reg add 29.7% cons** | 85 | 205 | 615 |
| **QP Reg add 54.3% cons** | 0 | 54 | 393 |
| **QP Reg add 100.0% cons** | 0 | 23 | 308 |

Table 10: Best number of violated R-matrix monotonicity constraints for the true Markov chain ordering, the Fiedler vector, the seriation QP in (6) and the semi-supervised seriation QP in (7) with varying numbers of pairwise orders specified. We observe the (randomly ordered) model covariance matrix (no noise), the sample covariance matrix with enough samples so the error is smaller than half of the spectral gap, then much fewer samples (Large noise).

|  | No noise | Noise within spectral gap | Large noise |
|---|---|---|---|
| **True** | 40145±0 | 40136±402 | 40588±4203 |
| **Spectral** | 40145±0 | 41417±1542 | 45765±5886 |
| **QP** | 43245±3548 | 43272±3081 | 45467±5008 |
| **QP Reg** | 43135±3448 | 43363±3073 | 45467±4946 |
| **QP Reg + 0.1%** | 45701±3700 | 46315±3352 | 46463±5582 |
| **QP Reg + 0.7%** | 47510±3870 | 47396±3800 | 49116±6349 |
| **QP Reg + 1.4%** | 48887±4029 | 48765±3933 | 49798±6040 |
| **QP Reg + 2.5%** | 47525±3705 | 48117±3805 | 50061±6053 |
| **QP Reg + 4.6%** | 47554±3070 | 47220±3166 | 49345±5687 |
| **QP Reg + 8.7%** | 45716±2652 | 46171±2559 | 47500±5606 |
| **QP Reg + 16.1%** | 43782±1985 | 43889±2545 | 45087±5140 |
| **QP Reg + 29.7%** | 41518±1148 | 41806±1376 | 42566±4423 |
| **QP Reg + 54.3%** | 40338±357 | 40409±500 | 41004±4230 |
| **QP Reg + 100.0%** | 40145±0 | 40136±402 | 40587±4201 |

Table 11: Median ± stdev. on objective value in problem (1) for the true Markov chain ordering, the Fiedler vector, the seriation QP in (6) and the semi-supervised seriation QP in (7) with varying numbers of pairwise orders specified. We observe the (randomly ordered) model covariance matrix (no noise), the sample covariance matrix with enough samples so the error is smaller than half of the spectral gap, then much fewer samples (Large noise).

|  | No noise | Noise within spectral gap | Large noise |
|---|---|---|---|
| **True** | 40145 | 39032 | 30820 |
| **Spectral** | 40145 | 39516 | 33805 |
| **QP** | 40145 | 39736 | 32935 |
| **QP Reg** | 40145 | 39736 | 32918 |
| **QP Reg add 0.1% cons** | 40220 | 40764 | 33158 |
| **QP Reg add 0.7% cons** | 41562 | 40642 | 32337 |
| **QP Reg add 1.4% cons** | 41554 | 42688 | 33472 |
| **QP Reg add 2.5% cons** | 41240 | 42012 | 33942 |
| **QP Reg add 4.6% cons** | 42397 | 41892 | 34176 |
| **QP Reg add 8.7% cons** | 41863 | 41453 | 34285 |
| **QP Reg add 16.1% cons** | 40558 | 40799 | 32346 |
| **QP Reg add 29.7% cons** | 40173 | 39523 | 31998 |
| **QP Reg add 54.3% cons** | 40145 | 39032 | 30820 |
| **QP Reg add 100.0% cons** | 40145 | 39032 | 30818 |

Table 12: Best objective value in problem (1) for the true Markov chain ordering, the Fiedler vector, the seriation QP in (6) and the semi-supervised seriation QP in (7) with varying numbers of pairwise orders specified. We observe the (randomly ordered) model covariance matrix (no noise), the sample covariance matrix with enough samples so the error is smaller than half of the spectral gap, then much fewer samples (Large noise).

Figure 3: The Hodson's Munsingen dataset: the first figure on the left has the order of the rows given by Kendall, the middle figure is the Fiedler solution, the figure on the right is the best QP solution from 100 experiments with different $Y$ (based on combinatorial objective).

Figure 4: Markov Chain experiments: the first figure on the left has the true order of the Markov chain, the middle figure is the Fiedler solution, the figure on the right is the best QP solution from 100 experiments with different $Y$ (based on combinatorial objective).