[Reviews · NeurIPS 2013]

Submitted by Assigned_Reviewer_6

This paper considers the problem of minimizing a quadratic form specified by a similarity matrix A over the class of permutation matrices. The authors first show that when the similarity matrix has a special structure (i.e. A is of the form A = C \circ C^T where C is a pre-Q matrix), then the quadratic form is minimized by the permutation that turns A into an R matrix. It is known that this permutation can be found by computing the Fiedler vector of the Laplacian matrix of A, so we can solve the optimization problem in polynomial time. The second main result is a regularized convex relaxation of the optimization problem in the noisy setting that also allows additional structural constraints. The authors proceed to demonstrate empirically that the convex relaxation is more robust to noise and can achieve a better solution (i.e. a lower objective value) than the manual or spectral solutions, given enough additional structural constraints.

This is a nicely written paper with interesting results, both theoretical and practical. The introduction and exposition are mostly clear, although the organization can be a bit improved. The two main results (the spectral solution in the combinatorial case and the convex relaxation in the noisy case) seem to be a bit disjointed. In Section 2, the connections between the different matrices, as well as the various preliminary results, can be confusing at times. Perhaps it would help if the main results (Proposition 2.6 and 2.11) are stated in the beginning of the section, or more emphasized than the other preliminary results.

Additional comments/questions:
* In Equation (2), why do we generalize \pi to y_\pi? It seems the extent of generality considered in this paper is when y is an affine transform of {1,…,n}, in which case (2) is clearly equivalent to problem (1). Can we say anything about more general y? Otherwise, it might help to stick with (1) to minimize notation and confusion.
* Although this has been stated in the introduction, it may help to state the problem of seriation mathematically in Section 2 (in terms of permuting A into an R matrix), so that the connection with the combinatorial problem becomes more explicit.
* L169-170 in the proof of Proposition 2.6 ("all monotonic subsets of y of a given length have the same (minimal) variance, attained by \Pi y") is not too clear to me. Perhaps it would help to write the problem (2) in terms of the Laplacian matrix (as in (3)) so it becomes more explicit what happens to the problem when we permute A.
* In Definition 2.8, what is the weight vector used in the subsequent discussion? Is it w = 1?
* L192: AB^T should be AB?
* In Proposition 2.11 (and 2.6), is the assumption that A = C \circ C^T with C pre-Q necessary? What about the case when A is a pre-R matrix, but not necessarily of the form A = C \circ C^T? And in practice, how can we check whether the conditions in Proposition 2.11 are satisfied?
* Is there anything we can say about the quality of the spectral solution when the assumptions of Proposition 2.11 do not hold exactly?
* I think Proposition 3.1 should be placed in Section 2.
* In the convex relaxation, how do we choose the values of \mu, p, and the perturbed Y in practice? Is the performance sensitive to the choice of these parameters?
* In the caption following Table 1, I think 24% should be 47.5%. I was also a bit confused about the claim that "the semi-supervised solution actually improves on both Kendall's manual solution and on the spectral ordering". Is this based on the combinatorial objective value? Is it clear that this is the correct metric to use to assess performance?
* Section 2 and 3 should make it more clear that some of the results are proved in the appendix, and state where in the appendix the results are proved.
* The additional experimental results in the appendix can also be organized or labelled better to make clear which application the results are for.
* Some of the plots can also benefit from a bit more explanation, such as what the ideal or best case is. For instance, what is the ideal behavior of Figure 3 in the appendix? Should the line be roughly diagonal? How do we compare the three matrices in Figure 5? And why are the two rightmost plots in Figure 2 have different formats?
* The second row, third column of Table 2 seems to be missing a value.
Summary: This is a nicely written paper with interesting results, both theoretical and practical. The introduction and exposition are mostly clear, although the organization can be a bit improved.

Submitted by Assigned_Reviewer_7

The authors first show a result involving decomposition of similarity matrices that allows for a connection between the seriation and 2-sum minimization problems on these matrices. They then show that the 2-sum minimization problem is polynomially solvable for similarity matrices coming from serial data. This result allows the authors to formulate seriation as a quadratic minimization problem over permutation matrices. They then produce a convex relaxation for this quadratic minimization problem. This relaxation appears to be more robust to noise than previous spectral or combinatorial techniques.

The seriation problem: we are given a similarity matrix between n variables. Assume variables can be ordered on a chain (i.e. along one dimension) where similarity corresponds to distance between them in the chain. The goal of seriation is to reconstruct the chain given unsorted, possibly noisy, similarity information. The seriation problem can be solved exactly by a spectral algorithm in the noiseless case; this paper proposes a convex relaxation to "improve the robustness of solutions" a noisy case. It aims to prove an equivalence between seriation and the combinatorial 2-sum problem over a class of similarity matrices. The combinatorial 2-sum problem is a quadratic minimization problem over permutations.

Overall, the paper was quite hard to read. There were lots of definitions and lemmas in a row without much motivation. While it is low on clarity, it seems to be original and possibly significant to those working on seriation problems.
Summary: The reviewer is unfamiliar with the area of seriation problems, and hence found it hard to judge the clarity of the paper - perhaps someone who was familiar with the definitions would have had an easier time browsing through this paper. Overall, I still found it well-written, and probably of significance to those working on noisy seriation problems, but it was hard to judge how much, so I give it the benefit of the doubt.

Submitted by Assigned_Reviewer_8

Summary:
This paper considers the seriation problem: given an unsorted and possibly noisy similarity matrix between a set of variables such that the variables can be ordered along a chain, where the similarity between variables decreases with their distance within this chain; reconstruct this linear ordering. This problem is known to be NP-complete, is related to the consecutive ones problem and also has a number of applications in archeology, gene sequencing, etc. Most of the earlier work is devoted to building convex relaxations based on semidefinite programs, which do not scale as the dimension increases. For a class of similarity matrices, the authors prove the equivalence between the seriation and the combinatorial 2-sum problem, a quadratic minimization problem over permutations, and show that in the noiseless setting, this nonconvex optimization problem can be solved exactly via a spectral algorithm, which is extended from [15]. In order to be able to handle noise and also impose additional structure for the seriation problem, the authors suggest a simple classical convex relaxation for permutation problems. They discuss application of two well-known methods, conditional gradient and accelareted gradient, for this solving the convex relaxation and provide promising numerical evidence.

Quality:
The equivalence between the seriation and the combinatorial 2-sum problem is new, to the best of my knowledge. Also the extensions to handle noise and impose additional structures for semi-supervised seriation problems are interesting. They also suggest a fast algorithm to project onto the set of doubly stochastic matrices, which is of independent interest. On the other hand the results seem to be a bit of elementary, in particular the convex relaxation is not novel, in fact it is obvious to see that it will be much weaker than the SDP relaxations suggested in the literature.

On page 5, lines 257-259, the authors claim that empirically, the results are stable even if the vector y is perturbed marginally. Having a theoretical justification of this observation could be of interest. They also suggest averaging the objective over several such small variations to improve its robustness to noise, i.e., eq (4). Having theoretical justifications for these would be nice.

On page 6, lines 270-275 a regularized nonconvex optimization problem, eq (5), is suggested, which is then modified in eq (6) to lead to a regularized convex problem. In an actual implementation, one needs to know the value of regularization parameter \mu for both of these problems. Perhaps some guidance can be offered in terms of how to select the best value for \mu.

Clarity:
The paper is mostly written in a clear fashion, with a reasonable logical structure. There are a few minor typos, grammatical errors, listed below. The motivation for the application example considered from the markov chain domain can be improved.

Originality & Significance:
To the best of my knowledge the approach taken is closely related with the reference [15], yet some of the results in this paper seem new and original. Moreover the topic can stipulate further interest in the broad NIPS community. Numerical evidence seems promising, besides being able to impose additional structure seems to come in as handy for gene sequencing applications, see discussion on page 8, lines 402-410.

Minor Issues:
a. On page 5, line 248: there is no “y” in eq (3), so “\Pi y” is confusing here
b. On page 7, lines 324-325: the wording of this sentence should be changed. Note that (7) not only involves constraints for doubly stochastic matrices but also structure inducing constraints, D^T\Pi g +\delta \leq 0.
c. Page 12, line 618: “numerical” should be “Numerical”
Summary: Overall I think this a well-written paper with some new insights and results, and the topic can stipulate further interest in the broad NIPS community and hence it can be accepted for publication.
Author Feedback

Author rebuttal: We would first like to thank the referees for their very constructive comments. We hope to clarify below some of the issues raised in the reports.

- "The two main results (the spectral solution in the combinatorial case and the convex relaxation in the noisy case) seem to be a bit disjointed."

The spectral result in [15] shows that the Fiedler vector of R matrices is monotonic but does not show that the solution of the 2-SUM problem is also monotonic for R matrices. Our main result in Section 2 is precisely to show the link between 2-SUM and seriation. Formulating seriation as a combinatorial problem (2-SUM) in Section 2 is what allows us to write a convex relaxation and solve semi-supervised problems derived from 2-SUM in Section 3. These additional structural constraints are not handled at all by the spectral solution.

- "Why do we generalize \pi to y_\pi?"

Our results are only stated for affine transforms of {1,...,n}. As commented, it would be nice to extend this result to perturbations of such vectors y. The results of section 2 show that seriation minimizes a weighted sum of variances of subsets of y, with the weights coming from the CUT representation of the matrix. This means that our results would still apply when both y and the CUT weights are not pathological, and we definitely plan to quantify this in future work.

Empirically, we have noticed that our algorithm still works for most monotonic vectors y. Moreover, in the noisy setting, some vectors y will give a tighter relaxation than others, and therefore averaging the objective function over several y's tends to improve performance. In practice we used p=4n, and chose y as sorted uniform random vectors in [0,1].

- "Perhaps some guidance can be offered in terms of how to select the best value for \mu." & "In the convex relaxation, how do we choose the values of \mu, p, and the perturbed Y in practice?"

Ideally, one would choose \mu large enough so that the optimal solutions of the problem are permutation matrices. However, this would make the problem non-convex and hence we could not guarantee finding a global optimum. Instead we suggest maintaining convexity by setting \mu=\lambda_2(L_A) \lambda_1(YY^T). This is the largest choice of \mu such that the optimization problem stays convex.

- "Is there anything we can say about the quality of the spectral solution when the assumptions of Proposition 2.11 do not hold exactly?"

The spectral solution in the noisy regime is guaranteed to be good in the perturbative regime where the “spectral gap” is small, that is when (lambda(3)-lambda(2))/2 \leq norm(L_A-L_A_noisy), where L_A is the laplacian of the pre-R matrix A (no noise, satisfying 2.11), L_A_noisy is the laplacian of the matrix A with some noise, lambda(2) and lambda(3) are respectively the second and third smallest eigenvalues of L_A. The numerical experiments on Markov chains test the performance of the spectral solution both inside and outside of this regime. In most other cases however, we do not have access to L_A and the spectral gap cannot be computed.

- "In Proposition 2.11 (and 2.6), is the assumption that A = C \circ C^T with C pre-Q necessary?"

This condition is required to get the CUT decomposition which is the key mechanism in our proof. We believe it can be relaxed, but we do not know how to show this yet. In practice, since the square of *any* R matrix can be written as C \circ C^T with C pre-Q, this hypothesis is somewhat harmless.

- "More explanations on the figures concerning gene sequencing (figure 2 p.8)"

If perfectly reordered, the graph of true read position versus recovered read position should be a monotonic line. The fact that we see a “zigzag” in the Fiedler plot shows that the spectral order did not recover the true position of the reads. This is mostly due to repeats in the original DNA sequence. In the rightmost plot of figure 3, we aggregate reads reordered by the Fielder vector into subsequences called “contigs”. This significantly shrinks the problem and these contigs are then themselves reordered using the semi supervised QP relaxation (with mate pairs information).

- "There were lots of definitions and lemmas in a row without much motivation. While it is low on clarity, it seems to be original and possibly significant to those working on seriation problems"

Space limitations severely constrained the amount of detail in the first part of the paper, which partly explains its relatively abrupt format. We really hope to significantly improve this in the revision and in the journal version.